# The role of heat flux-temperature covariance in the evolution of weather systems

Andrea Marcheggiani[1] and Maarten H.P. Ambaum[1]

[1]Department of Meteorology, University of Reading, Reading, UK

**Correspondence:** Andrea Marcheggiani (a.marcheggiani@pgr.reading.ac.uk)

**Abstract.** Local diabatic heating and temperature anomaly fields need to be positively correlated for the diabatic heating to maintain a circulation against dissipation. Here we quantify the thermodynamic contribution of local air–sea heat exchange on the evolution of weather systems using an index of the spatial covariance between heat flux at the air-sea interface and air temperature at 850 hPa upstream of the North Atlantic storm track, corresponding with the Gulf Stream extension region. The index is found to be almost exclusively negative, indicating that the air–sea heat fluxes locally act as a sink on potential energy. It features bursts of high activity alternating with longer periods of lower activity. The characteristics of these high index bursts are elucidated through composite analysis and the mechanisms are investigated in a phase space spanned by two different index components. It is found that the negative peaks in the index correspond with thermodynamic activity triggered by the passage of a weather system over a spatially variable sea-surface temperature field; our results indicate that most of this thermodynamically active heat exchange is realised within the cold sector of the weather systems.

## 1 Introduction

In the Northern Hemisphere, storm tracks have a limited longitudinal extent and are located mainly off the eastern coasts of mid-latitude Asia and North America. This is the case from a Eulerian (Blackmon et al., 1977) as well as a Lagrangian (Hoskins and Hodges, 2002) perspective.

Hoskins and Valdes (1990) emphasise the local Eady growth rate, the *baroclinicity*, as the dynamically relevant variable to determine the geographical structure of storm tracks. Ambaum and Novak (2014) point out the relevance of baroclinicity in describing the temporal structure of storm tracks. They define a two-variable model which combines local baroclinicity and meridional eddy heat fluxes in a nonlinear oscillator and subsequently Novak et al. (2015) make use of it to explain regime transitions of the mid-latitude eddy-driven jet stream, which had been previously observed by Franzke et al. (2011). In particular, Novak et al. (2015) found that oscillations in baroclinicity and heat flux lead to variability in eddy anisotropy, which could then be associated with a major change in the dominant type of wave breaking (Hoskins et al., 1983), consequently affecting the jet stream latitudinal position, as is also observed in idealised experiments (Rivière, 2009; Orlanski, 2003).

Meridional heat fluxes can be interpreted as an indicator for the conversion of mean-flow to eddy available potential energy in the Lorenz energy cycle (Lorenz, 1955). Meridional and vertical heat fluxes act as conversion terms across different types of energy reservoirs, whereas surface heat fluxes are associated with generation and dissipation of available potential energy.

Global estimates of these terms have been computed (Peixoto and Oort, 1992) and were used to identify the direction of energy flow within the Lorenz energy cycle. Novak et al. (2017) demonstrate that the dynamical relationship between storm track intensity and available potential energy as measured by baroclinicity can be described by a predator–prey relationship, whereby storm tracks can be thought of as *feeding* on baroclinicity.

The generation of eddy available potential energy in the Lorenz energy cycle is described analytically by a term which is proportional to the covariance between local heating and temperature (Lorenz, 1955; Peixoto and Oort, 1992; James, 1995). This term has been estimated to be positive globally (Oort, 1964; Oort and Peixoto, 1974; Ulbrich and Speth, 1991; Li et al., 2007; Marques et al., 2009), suggesting that diabatic processes are acting as a source of energy in storm development. However, this picture changes when we focus on the contribution of transient eddies, which correspond to synoptic scale weather systems,
to eddy available potential energy. Ulbrich and Speth (1991) provided a first estimate of the negative contribution of diabatic processes upon transient eddy energy, and more recent studies have further shown that the total diabatic generation of transient eddy potential energy is largely negative across the majority of the Northern Hemisphere's mid-latitudes (Chang et al., 2002), with diabatic heating damping transient eddy evolution, particularly during the winter season (Chang and Zurita-Gotor, 2007).

Diabatic processes at the surface, such as sensible and latent heat fluxes, can amplify horizontal temperature gradients by
heating where it is warm and cooling where it is cold, which is linked to the generation of available potential energy. From a global perspective this is achieved by the global differential in radiative heat input. However, the local thermodynamic effects of latent and sensible heat fluxes are much less clear: upward air–sea heat fluxes typically may be expected to coincide with a cooler local atmosphere, suggesting a negative contribution to the local potential energy budget.

The importance played by sea surface temperature (SST) fronts in forcing surface air temperature gradients through dif-
ferential sensible heating across the SST front has been highlighted in a series of studies (Nakamura et al., 2008; Hotta and Nakamura, 2011). This mechanism, called *oceanic baroclinic adjustment*, was shown to be essential for the maintenance of strong near-surface baroclinicity, which anchors the climatological storm track.

Chang et al. (2002), using a dataset composed by Januaries from 1980 to 1993, described the contributions of the different components of diabatic heating to eddy available potential energy and showed that latent and sensible heating can have different
effects on the potential energy budget. In particular, sensible heat flux was shown to have a strong local effect of relaxing the lower troposphere towards the underlying sea surface (Chang et al., 2002; Swanson and Pierrehumbert, 1997; Hotta and Nakamura, 2011), while latent heating was not necessarily linked to local provision of heat input, because condensation may happen at a different location.

The intensity and sign of surface heat fluxes are typically computed from the near surface atmospheric conditions, hence
their covariation with higher layers of the atmosphere is non-trivial and it can have an effect on the evolution of weather systems. The aim of this study is to identify and describe this local thermodynamic effect of air–sea heat fluxes.

In particular, we examine how synoptic heat fluxes contribute to enhancing or depleting the local synoptic temperature variance in the lower troposphere. This local temperature variance is part of the global available potential energy integral in the standard Lorenz energy cycle. Therefore, we construct a hybrid framework where we can consider the spatial covariance
between anomalous heat flux and temperature fields as a measure of the local contribution to diabatic generation or destruction

of available potential energy. We focus on the link with synoptic storm evolution by using time anomalies for all atmospheric fields as deviations from a synoptic-timescale mean.

This article is structured as follows: Section 2 briefly summarises the Lorenz energy cycle and the approach we take in our study. Section 3 introduces heat flux-temperature spatial covariance and examines its main features through the definition of an index. Section 4 investigates the driving mechanisms of the index previously introduced. Finally, in the final section results are summarised and discussed.

## 2 Lorenz Energy Cycle and flux–temperature covariance

Available potential energy can be generated globally through differential heating which amplifies the global meridional temperature gradient and gives the troposphere in the mid-latitudes a baroclinic structure favourable to the growth of extra-tropical weather systems (Peixoto and Oort, 1992). In the Lorenz energy cycle (Lorenz, 1955) the interaction between different types of energy reservoirs is represented by conversion terms while surface heat exchange appears in energy generation and dissipation terms. Global estimates of these terms have been computed (Oort, 1964; Oort and Peixoto, 1974; Ulbrich and Speth, 1991; Li et al., 2007; Marques et al., 2009) and they are found to differ not only in time from seasonal to inter-annual scales, but also depending on the type of data variability considered, be it purely temporal, spatial or a combinations of these. For example, Oort (1964) found that generation of eddy available potential energy was negative in a spatial domain, whereas in a mixed space-time domain this was found to be positive. Ulbrich and Speth (1991) further decomposed eddy energy into stationary and transient components and estimated their generation to be positive for the stationary and negative for the transient component. Their estimates were based on January and July from 1980 to 1986 and shared the same signs, although with a difference in magnitude.

The generation and dissipation terms have normally been estimated as residuals in the main balance equations, as data for their direct computation typically were not archived. Global estimates normally suggest a positive generation of eddy available potential energy, which would involve heating of warm and cooling of cold air masses. Locally, however, model experiments with simplified climate models, where diabatic heating is determined as a relaxation of the temperature field, show a negative generation of eddy potential energy, with diabatic effects damping eddy available potential energy. This is also supported in studies by Swanson and Pierrehumbert (1997) and Chang et al. (2002), where they highlighted the importance of lower tropospheric thermal adjustment on short timescales to the underlying sea surface.

Given that storm tracks are by definition the main reservoirs of eddy potential energy, this begs the question of whether diabatic effects in storm tracks actually help or hinder their development, as investigated by Hoskins and Valdes (1990) who envisaged that sensible heating of cold air masses actually decreases the energy of weather systems while latent heating helps in their intensification in the warm sectors.

Given that there are different formulations of available potential energy budgets with each giving different interpretations from the same data, we will not favour any particular formulation here. Instead, we take a hybrid approach: we use direct estimates of surface heat fluxes over the upstream sector of the North Atlantic storm track region and use it to estimate whether

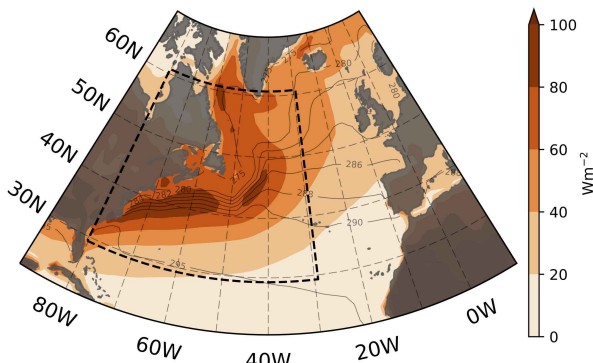

**Figure 1.** Temporal standard deviation of F (shading) and SST winter climatology (contours, every 2K from 280K to 290K, every 5K otherwise). The area within the dashed box (30–60°N, 30–79.5°W) corresponds to the region of the N. Atlantic considered in the next sections for the computation of spatial averages.

it can serve as a source or as a sink of spatial variance in temperature. Available potential energy is a global measure of such temperature variance. By defining a spatial covariance index between air–sea heat fluxes and lower atmospheric temperature we can quantify the extent to which the local heat fluxes help build available potential energy, or deplete it.

In particular, we consider the spatial covariance between time anomalies in instantaneous air–sea heat fluxes $F'$ and air temperature $T'$ at 850 hPa to define an area specific FT index,

$$\text{FT} = \langle F'^{*}T'^{*} \rangle = \langle (F' - \langle F' \rangle)(T' - \langle T' \rangle) \rangle = \langle F'T' \rangle - \langle F' \rangle \langle T' \rangle, \tag{1}$$

where primes denote time anomalies, angle brackets spatial averages over the area selected and stars deviations from this spatial average. The reasons behind the choice for the 850 hPa level as reference temperature are presented later in Section 3.

In order to concentrate on synoptic scale variability, time anomalies are defined as deviations from a running mean with a time window of 10 days (Athanasiadis and Ambaum, 2009). By removing a 10-day running mean in the construction of anomalies, we are filtering out lower-frequency variability, such as seasonal variations, which may otherwise dominate the spatial variance, and which describes different physical processes.

Data come from the European Centre for Medium-Range Weather Forecast (ECMWF) Re-Analysis Interim dataset (ERA-Interim, see Dee et al., 2011), restricting our attention to wintertime only (December to February, DJF), 6-hourly data from December 1979 to February 2019, for a total of 40 winters, interpolated onto a spatial grid with a resolution of 1.5° in both latitude and longitude. Instantaneous surface sensible heat fluxes have been utilised as a measure for heat exchange, $F$, which we define as positive if heat flows upwards from the ocean to the atmosphere.

Repeating our analysis with latent heat fluxes or the sum of latent and sensible heat fluxes did not substantially change the outcomes we report on here, although values depending on heat flux magnitude of course change. The fact that the analysis seems mostly independent of which flux is used, indicates that the space and time filtered fluxes have a broadly fixed Bowen ratio on synoptic time scales.

The FT index was calculated over the western North Atlantic, extending between $30° - 60°$N and $30° - 79.5°$W, masking out land grid points in order to concentrate on air–sea interaction only. The domain selected is shown in Fig. 1 and coincides with both the upstream region of the storm track and the Gulf Stream extension, where the largest SST variability is observed across different scales (e.g. large-scale meridional gradients and small-scale oceanic eddies). Neither the spatial resolution chosen nor the finest resolution available in ERA-Interim would allow for oceanic eddies to be fully resolved. However, given

that the computation of $F$ relies on $T$ at the surface and air temperature is assimilated from observations, their effect on $F$ at the resolved scales would be captured by the reanalysis system and they would still contribute some residual variance which is included in our analysis.

## 3   Temporal properties of the FT index

Figure 2 (top) shows the temporal behaviour of the FT index, Eq. 1, as defined for the upstream region of the North Atlantic

storm track.

    The index is found to be always negative and it features moderately frequent (strongest 5th percentile occurring once every 2–3 weeks) bursts of intense activity peaking at values down to almost $-1500$ Wm$^{-2}$K among periods of weaker activity during which the index fluctuates around values closer to zero, although still keeping its negative sign. This is reflected in the empirical distribution of the index values, plotted to the right of the index time series in Fig. 2, featuring large skewness and an

extended tail towards negative values, as well as a cut-ff for positive values.

    The empirical distributions for the local values of $F'^*$, $T'^*$ and $F'^*T'^*$ are shown in Fig. 2 (bottom left to right respectively). More than $9.5 \times 10^6$ data points across both the spatial and time domain were used, which allowed for the distributions in Fig. 2 to be examined without any sort of data filtering. These anomalies correspond to the anomalous fields constructed in order to calculate the index, which is the spatial average of $F'^*T'^*$.

The distribution for heat flux space-time anomalies is distinctively skewed towards positive values, whereas temperature anomalies follow more a Gaussian distribution. This is consistent with the different heat capacities of the atmosphere and the ocean, as the atmosphere is more easily heated by the ocean, while it takes both a longer time and a stronger vertical gradient in temperature for the atmosphere to flux heat into the ocean.

    The product of the local heat flux and temperature anomalies, on the other hand, shows an asymmetric distribution markedly

skewed towards negative values with a long negative tail, indicating strong local negative correlation between the two variables. There are however a substantial number of positive values of the local product. These positive values correspond to heat flowing from an anomalously cold sea-surface to an anomalously warm air mass (and vice versa). The FT index is the spatial average of this signal and it is found to be always negative.

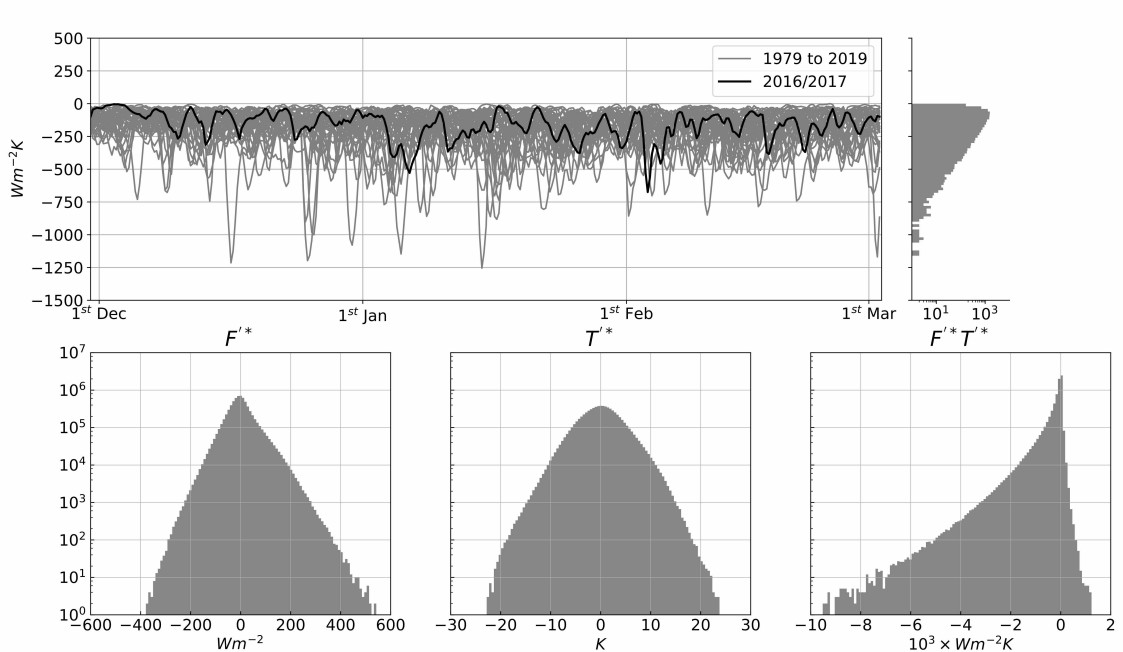

**Figure 2.** Top: Index time series computed over the upstream region of the N. Atlantic storm track (30–60°N, 30–79.5°W), spanning the full ERA-Interim time series (grey solid lines), highlighting a sample season (2016/2017 winter, solid black line); (right) empirical distribution of index values (semi-log scale). Bottom: Empirical distribution of instantaneous space-time anomalies in surface heat flux and temperature over the upstream region (semi-log scale).

The local product is most often negative given that the air–sea heat fluxes are parameterised in terms of the temperature difference between the sea surface and the lower atmosphere. However, high instability in the lowest layers of the troposphere could cause the local product to become positive, as air temperature at 850 hPa is not directly used in the computation of surface heat fluxes. Furthermore, the transfer coefficient is a non-trivial function of boundary layer properties, not directly linked to the temperature at 850 hPa. It is therefore a non-trivial result that the FT index is observed to be negative at all times.

The sporadic nature of the strong negative index values suggest a link with weather system activity, as observed in for example in Messori and Czaja (2013) and Ambaum and Novak (2014). Evidence for this link is shown in Fig. 3, where composites on the strongest and weakest FT index values are shown for mean sea level pressure, air temperature at 850 hPa and surface sensible heat flux. Strong FT index values (in the most negative 5th percentile) correspond to patterns associated with a low pressure system, with stronger than usual surface heat flux coinciding with cold air being advected from the American continent. Weak FT index (values in the top 5th percentile) correspond instead to inhibited storm activity, with weaker surface heat flux consistent with a pressure pattern which leads to weakened low level westerlies.

We chose $T$ at 850 hPa as it is not directly involved in the computation of $F$ and, therefore, its covariation with $F$ is non-trivial and entails more information about development of the synoptic systems. Use of surface air temperature ($T$ at 2-metre height) would serve to emphasise the strong interlink between temperature and surface heat fluxes, the computation of which directly involves $T$ at the surface. In fact, covariances appear to be *weaker* when considering $T$ at the surface, as temperature variance is higher at the 850 hPa level and, indeed, the distribution for correlation between $F$ and $T$ at the surface (not shown) is slightly shifted towards stronger values, while correlation between $F$ and $T$ at 850 hPa (Fig. 2) features a longer tail towards weak values. Composites for 2-metre temperature for weak and strong FT index (defined as before using $T$ at 850hPa) are found to be similar to composites for $T$ at 850 hPa (Fig. 3b,e) with slightly weaker anomaly values. This is likely caused by contribution from uncorrelated boundary layer dynamics to surface temperature and the suppression of correlated variance by relaxation to the underlying SSTs.

Lagged composites centred on extreme events were also computed for mean sea level pressure, air temperature and precipitation rates, both convective and large-scale (as available from ERA-Interim, Dee et al., 2011), though not shown for the sake of conciseness. Between four and three days before the peak intensity in the FT index is reached, a low pressure system was observed entering the spatial domain, then intensifying at the FT index peak and finally decaying within synoptic time scales (three–four days).

The intensification and decay phases observed in the lagged composites partially derives from a gain or loss of signal due to averaging of several different kinds of events, especially at longer lags. However, the decay phase was observed to be relatively rapid compared to the intensification phase, as weather patterns leading to the peak were observed to last longer than those following the peak. This asymmetry between the initial and final stages of the FT index intensification is consistent with the idea that a strong negative FT index indicates a thermodynamic sink on the system.

The time average of $F'^{*}T'^{*}$, shown in Fig. 4, provides us with a picture of where the spatial covariance between $F'$ and $T'$ is realised within the spatial domain under consideration. This is found to peak along the Gulf Stream, where also the largest $F$ time variance is observed (compare with Fig. 1), thus advocating for the importance of SST variability in shaping the $F' - T'$ spatial covariance.

Note in addition that the FT index is a measure of spatial variability and concurrent positive or negative anomalies in $F$ and $T$ do not necessarily correspond to stronger or weaker values compared to climatology, rather it would indicate a stronger or weaker intensity compared to both the surrounding area and the previous and following 5 days. We also found weaker negative FT index values to be indicative primarily of diminished storm activity, as Fig. 3 shows. Hence, it is reasonable to interpret any positive instances or moderately negative values as indicative of a relatively weak heat exchange, in the quiescent period between storm systems.

## 4 Phase-space properties of the FT index

We expect the FT index to be associated with variations in storm track properties. In order to get a clear picture of these associations we will employ a phase space kernel averaging technique.

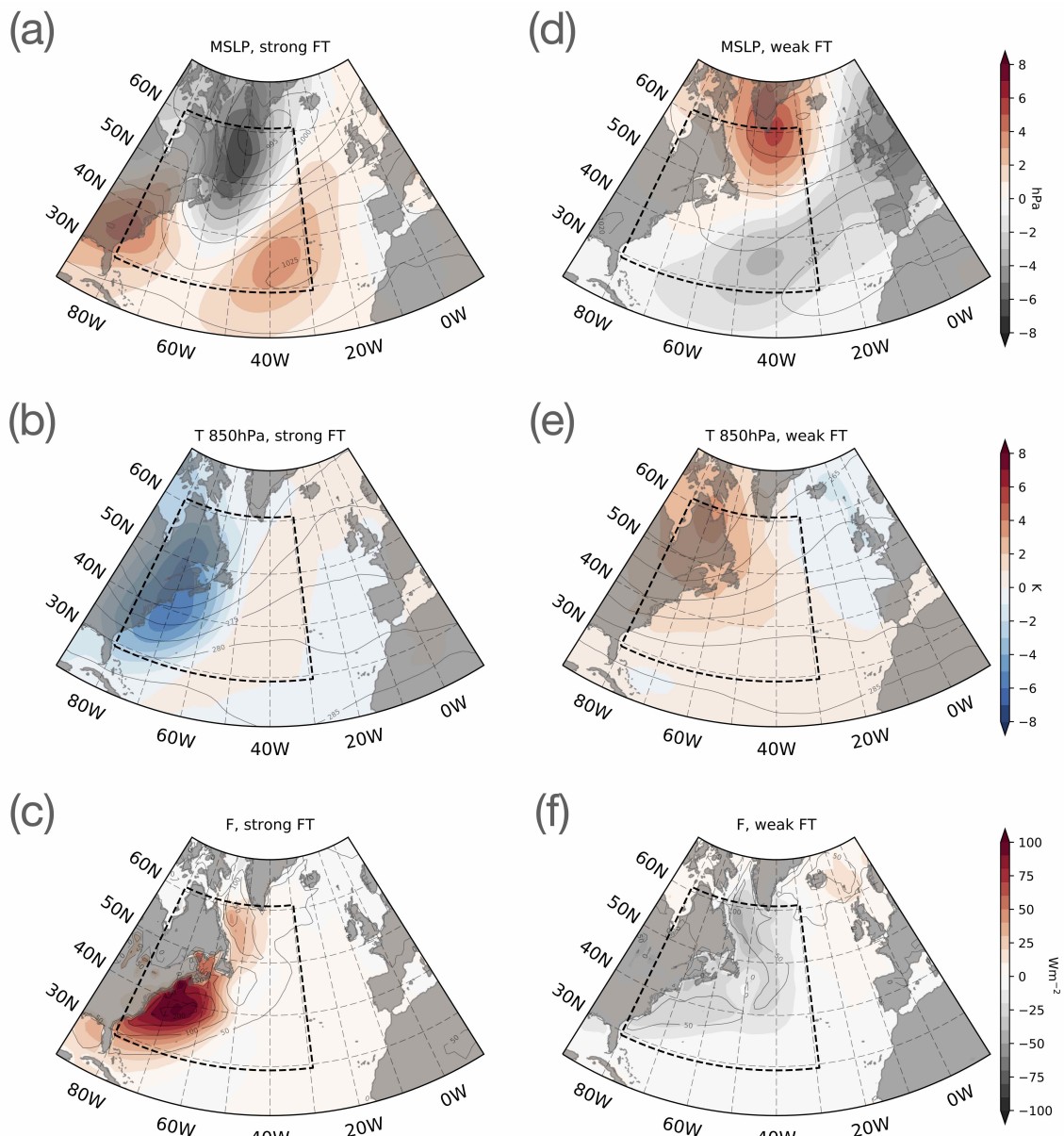

**Figure 3.** Composites on strongest (a–c) and weakest (d–f) FT index values (top and bottom 5th percentiles) for mean sea level pressure (a,d), air temperature at 850 hPa (b,e) and surface sensible heat flux (c,f). Contours and colour shadings represent, respectively, composites and their difference from winter climatology; dashed boxes indicate the spatial domain where the FT index is defined.

The phase space is spanned by two variables. Any quantity can be kernel-averaged at any point in the phase space. We thus obtain a picture of how the quantity will depend on the two variables spanning the phase space.

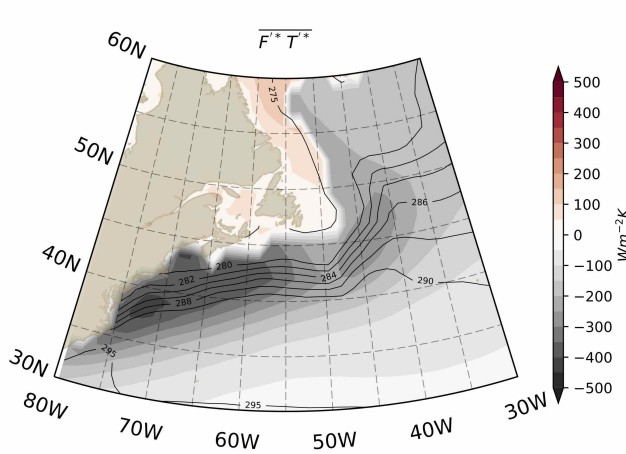

**Figure 4.** Wintertime (DJF, 1979-2019) mean of the product between time-space anomalies in $F$ and $T$ over the spatial domain selected for our study (shading) and wintertime SST climatology (contours, every 2K from 280K to 290K, every 5K otherwise).

A particularly interesting quantity to represent in such phase space is the tendency of the variables that span the phases space. In this way we can construct a flow in the phase space, representing the kernel averaged tendencies in the data.

The technical details of constructing the phase space averages and tendencies are described in Novak et al. (2017). They constructed a two-dimensional phase space where they were able to identify a predator-prey relationship between meridional heat fluxes and mean baroclinicity respectively, as these were used as coordinates in the phase space. Results may vary somewhat according to kernel size chosen, though in our study the results were observed to be broadly independent of the size of the kernel used for all reasonable size choices (not shown).

We start our analysis by constructing a phase space spanned by the FT index and baroclinicity spatially averaged across our chosen N. Atlantic storm track domain. Following Hoskins and Valdes (1990) and Ambaum and Novak (2014), we calculated baroclinicity as the Eady growth rate maximum at 750hPa (Ambaum and Novak, 2014, see Eq. 10), taking a linear approximation for the vertical gradient in zonal wind between the 650hPa and 850hPa levels. The kernel averaged phase tendencies for the FT index and mean baroclinicity are shown in Fig. 5.

We find that the circulation in the FT–baroclinicity phase space lies entirely on the negative side of the FT index axis and it is in the anti-clockwise direction. (The few trajectories crossing into the positive FT index region are due to kernel smoothing.) The phase portrait indicates that mean baroclinicity starts becoming depleted once the FT index has strengthened enough and it recovers only at lower FT index values, which is consistent with results of composite analysis, whereby baroclinicity was found to decrease during extreme events in the FT index (not shown). The observed baroclinicity depletion could be linked to the growth of baroclinic waves happening at the same time as the FT index increases, therefore care should be taken in inferring causality. Nonetheless, while air–sea fluxes at low frequencies maintain and anchor the high baroclinicity regions (Hotta and

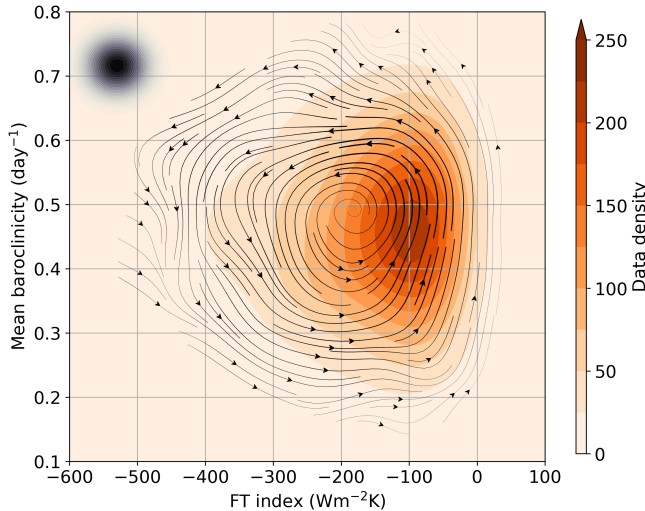

**Figure 5.** Kernel-averaged circulation in the FT index-mean baroclinicity phase space. Streamlines correspond to kernel-averaged rates of change in FT and baroclinicity (line thickness proportional to phase speed, plotted where data density is larger than 10). Colour shading represents kernel-smoothed data density. The size of the averaging Gaussian filter is indicated by the black-shaded dot in the upper-left corner.

210    Nakamura, 2011), our analysis is consistent with the picture that, at higher frequencies locally in time and space, these heat fluxes damp the synoptic-scale temperature variance, as the negative FT index acts as a measure of both eddy amplitude and of how air–sea heat fluxes might erode local temperature gradients (i.e. baroclinicity).

These results do not contradict the findings by Hotta and Nakamura (2011) on the role of sensible heating at the surface in restoring baroclinicity and are actually complementary to them. In fact, the spatial variance of the fluxes includes contributions

215    also from the north-south gradient of SSTs over the oceanic front. This is consistent with the mechanism discussed in Ambaum and Novak (2014) and Novak et al. (2017) where the authors highlight the role that eddies play in temporarily depleting the baroclinicity in a predator–prey like relationship. This relationship is really an instance of the nonlinear life-cycle of midlatitude eddies where eddy activity locally depletes the meridional temperature gradient in the atmosphere. (In the older literature this quasi-periodic predator–prey relationship would have been described as an index cycle.) However, this does not contradict

220    the fact that high eddy activity on average must be geographically associated with high baroclinicity, as argued by Hotta and Nakamura (2011), Ambaum and Novak (2014) and elucidated also in earlier studies by Swanson and Pierrehumbert (1997) and Hoskins and Valdes (1990).

Our analysis suggests that the flux-temperature spatial covariance plays an important role in the budget for mean baroclinicity and, more generally, for available potential energy (Ambaum and Novak, 2014), alluding to the existence of a link between

225    any driving mechanism behind the FT index and storm evolution. In fact, our result shows that the FT index is a good measure of processes that deplete baroclinicity.

The FT index can be decomposed into the product of flux-temperature spatial correlation and spatial standard deviations in flux and temperature,

$$\text{FT index} = \text{cov}(F', T') \equiv \text{corr}(F', T')\, \sigma(F')\, \sigma(T'). \tag{2}$$

This suggests we can also use spatial standard deviations in $F'$ and $T'$ as coordinates of the phase space where trajectories traced by the index components would represent its evolution across the various components of the index.

The occurrence of strong index values can be explained by increasing variance in either heat flux or temperature, or anomalously strong correlations between the two variables. Another possibility is of course that a combination of any of these three factors produces strong index events. This question of magnitude driven or phase driven index extremes is analogous to that presented in Messori and Czaja (2013) for meridional heat transport and our phase space analysis provides a novel viewpoint of the phenomenon.

Figure 6a shows the result from kernel-averaging in a phase space spanned by the variances in heat flux and air temperature. Here streamlines indicate the phase space mean trajectories and their thickness is proportional to the phase speed, while the shading represents the typical value of the FT index at each point in the phase space as resulting from kernel-averaging. Regions in the phase space where data is scarce (less than 10 and 1 data points respectively for streamlines and FT index value) are hidden as kernel-averages there are not representative of the local value of the variable.

The trajectories traced by the FT index components are found, on average, to be oscillating between low and high values of the index, which is consistent with the behaviour observed in the time series and shows that stronger index values are associated with larger variances in $F'$ and $T'$. The trajectories are also observed to be oscillating between weak and strong $F'$-$T'$ spatial correlation, as shown by spatial correlation phase tendencies illustrated in Fig. 6b.

Taking a closer look at the relationship between spatial correlation and standard deviations in $F'$ and $T'$, these appear to be growing concurrently. This can be deduced by inspecting Fig. 6b, as spatial correlation is observed to increase together with the product of spatial standard deviations in $F'$ and $T'$, which is represented by grey contours.

In Fig. 7, spatial correlation is plotted against the product of standard deviations in $F'$ and $T'$ using values from Fig. 6b (dark-grey dots) and then compared with raw data (light-grey dots) in order to exclude it being an artefact of kernel-averaging. Spatial correlation and variances are found to be in an almost log-linear relationship, with both phase space data and raw data indicating an increase in correlation strongly linked to an increase in variances.

These results suggest that the observed bursts in flux-temperature spatial covariance are neither exclusively phase-driven nor exclusively magnitude-driven. Both high flux and air temperature spatial variability and correlation are characteristic features of the bursts. We conclude that both strength and correlation in spatial variability are equally fundamental to the build up of flux-temperature spatial covariance.

It is not clear why the correlation between the two variables increases so markedly with their variability. The simple model of flux being essentially proportional to the temperature at 850 hPa (minus the SST) would not exhibit such a behaviour.

The simultaneous growth of correlation and variance is a non-trivial result and it suggests further research into assessing whether this is a general feature of the relationship between flux and lower atmospheric temperature or if it is limited to

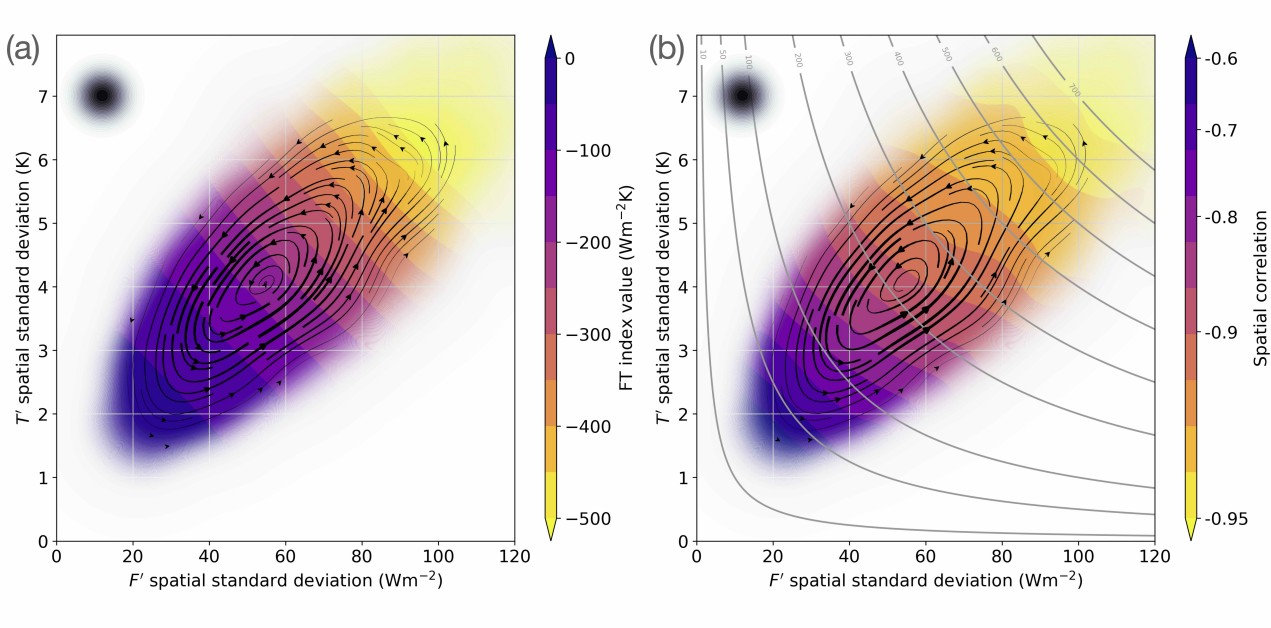

**Figure 6.** Kernel-averaged circulation in the $F'$-$T'$ spatial standard deviations phase space. Streamlines correspond with kernel-averaged trajectories traced by the product of spatial standard deviations (line thickness proportional to phase speed, plotted where data density is larger than 10). Shading represent values of the FT index value (panel a) and FT spatial correlation (panel b). Grey contours in panel (b), drawn at 10 $\mathrm{Wm^{-2}K}$, 50 $\mathrm{Wm^{-2}K}$, 100 $\mathrm{Wm^{-2}K}$ and then every 100 $\mathrm{Wm^{-2}K}$, indicate the product of spatial standard deviations. The size of the averaging Gaussian filter is indicated by the black-shaded dot in the upper-left corner.

spatial variability dynamics or to the specific timescales considered. This would go beyond the scope of the present paper, but preliminary analysis indicates that the increase of correlation with variance may be a more generic property of the relationship between air–sea flux and lower atmosphere temperature.

 The kernel-averaged trajectories in the phase space are organised in concentric ellipses, which suggests that the evolution
265 of the FT index is cyclical in nature. By computing the average phase speed at which the trajectories are traced, it is estimated that it takes between 4 and 6 days for the FT index to go round a full cycle (see Fig. 10a for a sample trajectory). This time frame (4-6 days) falls within the range of synoptic timescales, consistent with the idea that the index is closely linked to the evolution of a storm system.

 We then notice that the observed circulation spins in an anticlockwise direction. This indicates that the spatial variability in
270 $F'$ leads in time on the spatial variability in $T'$, as can be seen by following any of the trajectories starting from weak index values.

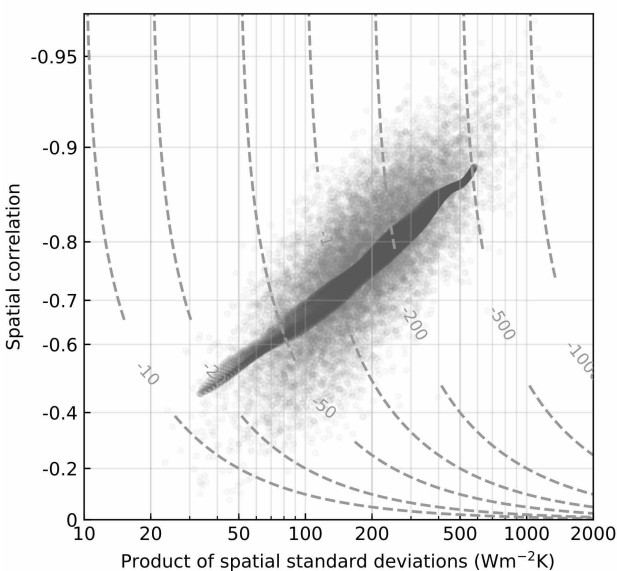

**Figure 7.** Scatter plot of $F'$-$T'$ spatial correlation against the product of $F'$ and $T'$ standard deviations using kernel-averaged data points from phase portrait (dark shading) and raw data (grey dots); grey contours (Wm$^{-2}$K) indicate FT index value.

This is somewhat counter-intuitive. A possible explanation is that this effect could be caused by the advection of cold air with a more spatially uniform temperature pattern over the Gulf Stream extension region, which features a much more spatially variable temperature field. SST spatial variability would then trigger heat flux spatial variance and subsequently lead to temperature variance generation. In the case of a weather system, the effect of surface heat fluxes would be that of eroding the spatial temperature variance by damping the cold sector temperature anomaly, while the warm sector is less affected by this coupling with the surface. Kernel averages for strong and weak spatial standard deviations in $F'$ and $T'$ (not shown) were found to be able to reproduce the same spatial structures that are found by compositing on extreme values (Fig. 3), which further supports the idea of the cold sector playing a primary role in the evolution of the FT index.

Further evidence to the importance of the cold sector is gathered by inspecting phase tendencies of $F$ and $T$.

Figure 8 shows phase tendencies for spatial-mean heat flux $F$ and air temperature $T$. We find that the growing phase of the FT index coincides with a decrease in mean $T$ and a concomitant increase in mean $F$. A decay phase then follows, characterised by the opposite trends.

Heat flux anomalies range from $-20$ Wm$^{-2}$K in the decay phase up to $40$ Wm$^{-2}$K in the growing phase, while air temperature anomalies stretch between $-2$ K and $2$ K respectively. The standard deviations in time of spatial-mean $F$ and $T$ are respectively $23.3$ Wm$^{-2}$K and $2.2$ K, suggesting that these signals do not arise exclusively from random fluctuations and thus providing our results with robustness.

Phase tendencies in Fig. 8 may be explained by relating the growing and decaying phases to an increased dominance of the cold sector of weather systems in the former, while the warm sector influences the latter. This would be in agreement with composite analyses for convective precipitation (not shown), which showed a precipitation band coinciding with the cold front as identifiable from the air temperature composite.

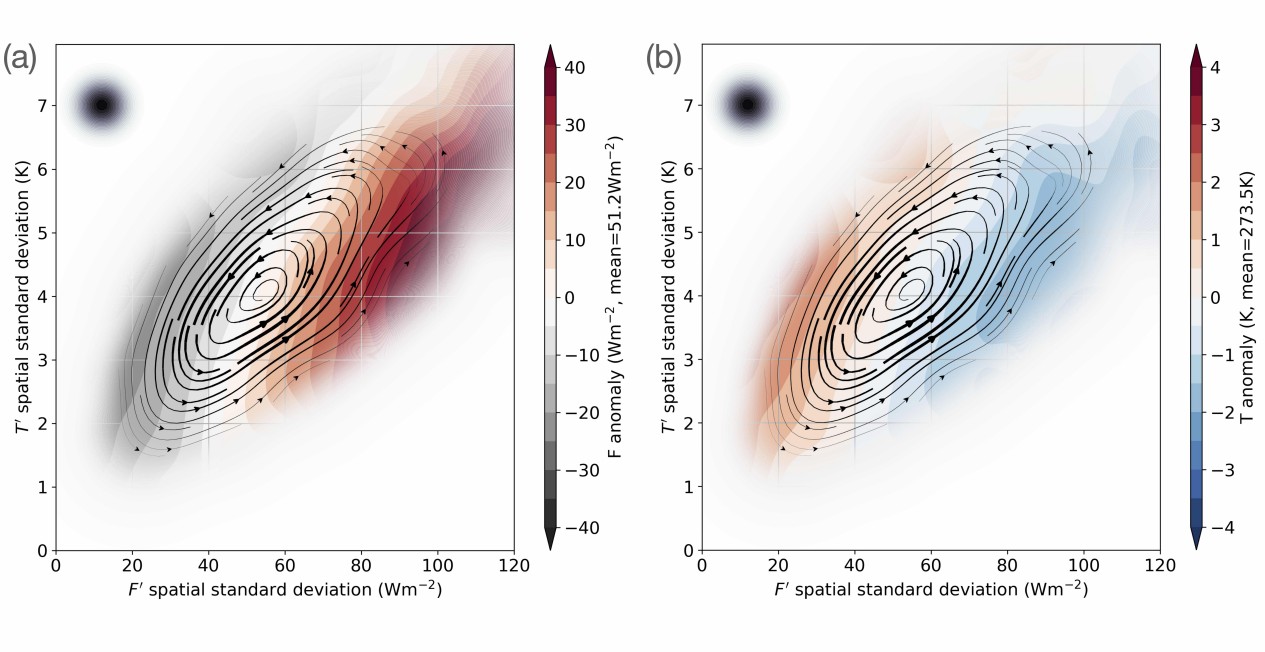

**Figure 8.** Phase tendencies for spatial-mean $F$ (a) and $T$ (b). Shading represent difference between phase tendency and the mean value of $F$ and $T$, as reported next to each colour bar. Streamlines as in Fig. 6. The size of the averaging Gaussian filter is indicated by the black-shaded dot in the upper-left corner.

Further evidence to the importance of the cold and warm sectors in the evolution of the FT index can be found in a more detailed analysis of the index dynamics in the phase space. A closed trajectory in the phase space is chosen by selecting a line of constant value of the stream function which was computed to draw the streamlines shown in the phase portraits. The selected closed trajectory is illustrated in Fig. 9 and a complete revolution takes about 5 days. It crosses regions of high data density so that it corresponds to a large number of unfiltered trajectories (i.e. not kernel-averaged) and thus presents a statistically robust picture.

The evolution in time of the potential temperature vertical profiles along the closed trajectory is portrayed in Fig. 10a, which shows the difference between the kernel-average and climatology along the closed trajectory shown in Fig. 9. The kernel-averaged mean boundary layer height is also plotted, together with the climatological mean.

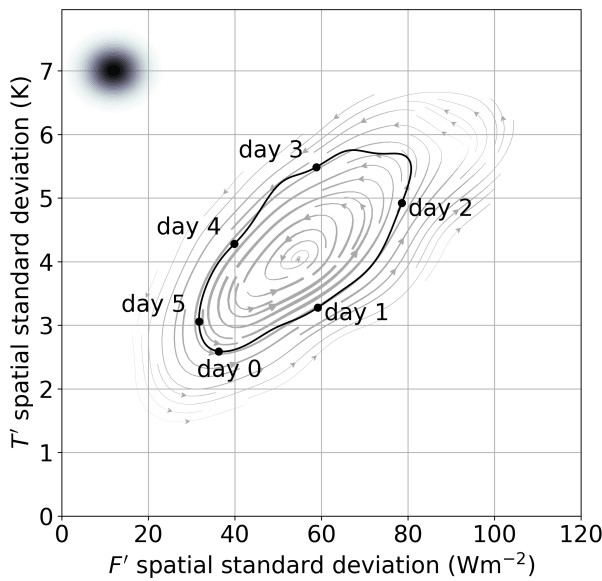

**Figure 9.** Closed trajectory in the phase space of $F'$ and $T'$ spatial standard deviations chosen for the computation of phase tendencies evolution. Black dots along trajectory indicate time duration in days of each section. The size of the averaging Gaussian filter is indicated by the black-shaded dot in the upper-left corner.

The cold and warm phases are characterised respectively by deeper and shallower atmospheric boundary layer. This is compatible with the idea that the growing phase corresponds to the advection of the cold sector into the spatial domain over a warmer SST leading to instability and convective heat fluxes. Furthermore, we inspected the time evolution of the anomalous wind direction closer to the surface, at the 950hPa level, as shown in Fig. 10b, and we found it to be consistent with cold air advection in the first half of the cycle, with a north-westerly anomaly backing to a south-easterly wind anomaly and warm air advection in the second half of the cycle. The anomalous wind was computed by removing the climatological mean wind, which is broadly westerly as expected along the storm track.

The strongest temperature anomalies are observed in the lower layers of the troposphere, which is symptomatic of the close relationship between the FT index and surface heat exchange, as per definition of the index itself. A tilt in the anomalous temperature profile is observed, especially in the cooling phase where the temperature anomalies are largest, as the cold sector moves across the spatial domain. It is not clear whether this tilt can be related to baroclinic lifecycle. Lim and Wallace (1991) diagnosed a weak forward tilt of temperature also at lower levels, as it must be for growing waves (Hoskins and Heckley, 1981), though substantially less than the westward tilt of geopotential. The magnitude of the tilt is hard to compare to our results as the $x$-axis in Fig. 10 maps onto time in a non-trivial way. The stronger tilt/lag of temperature at upper levels that we find is

not consistent with observations or expected from theory of idealised life cycles, where in the lower stratosphere at least, the tilt/lag is expected to reverse, as suggested in Lim and Wallace (1991) and Hoskins and Heckley (1981).

The warm phase coincides with a shallower boundary layer, as warm air is advected over the cold side of the SST front, which results in a more stable atmospheric boundary layer and weaker heat exchange. Indeed, the sea surface does not reach temperatures as low as in the preceding cold sector, hence it does not interact as strongly as in the cold phase and this could
explain the rapid decay of the heat flux–temperature spatial covariance.

We find that these results are not sensitive to the choice of the specific closed phase space trajectory (not shown).

The heat exchange within a cold sector arguably plays a primary role in driving the FT index. The phase tendency of the area fraction of the spatial domain occupied by the cold sector, shown in Fig. 11, illustrates this further. To estimate the area fraction, we utilise a a diagnostic based upon potential vorticity at the 95kPa level as proposed in a study by Vannière et al.
(2016), where it is shown that the cold sector is characterised by a negative potential vorticity signature which proved to be effective as a diagnostic through the comparison with more traditional indicators of the cold sector of extra-tropical weather systems.

In the strengthening phase of the FT index life cycle, the extent of the cold sector almost doubles from about 20% to almost 40% of the domain. This suggests that air–sea heat exchange in the cold sector may have significant effects on storm evolution,
in particular by driving the depletion of the baroclinicity over the domain, in accordance with Fig. 5. This appears to be in contradiction with earlier findings in Vannière et al. (2017), where it was suggested that baroclinicity is mainly restored in the cold sector.

Looking at specific events in the FT index, we find that surface heat flux and SST fields are well correlated, especially over warmer sea surfaces. SSTs over the Gulf Stream extension region are indeed characterised by higher spatial variability than air
temperatures due to the presence of both a strong SST front linked to the Gulf Stream and mesoscale oceanic eddies.

Oceanic mesoscale eddies have been shown to play a decisive role in shaping the North Atlantic storm track as they support stronger storm growth rates, making their representation essential for a better description of the storm track (Ma et al., 2017; Zhang et al., 2019). In particular, Foussard et al. (2019) examined the effect of oceanic eddies on storm tracks through an idealised experiment focused on the mid-latitudes, observing a poleward shift of storm trajectories compared to simulations in
which mesoscale eddies are removed, as found by Ma et al. (2017) in more realistic simulations for the North Pacific. Foussard et al. (2019) noticed also a larger sensitivity of the atmosphere to positive than to negative anomalies in SST, as the former correspond to a stronger temperature gradient at the air–sea interface.

Vertical motions associated with along-frontal flow could be expected to play a significant role in driving the FT index. However, our data would suggest that their role is not dominant.
In fact, if the FT index reached its highest point when frontal circulation is strongest we would expect the mean temperature to be around average, as a front is associated with both anomalously warm and cold air masses. Instead, we find that the FT index peaks when the area mean temperature is coldest, and when the cold sector area is largest. Furthermore, we find that temperature variance peaks when the FT index (i.e. FT co-variance) also peaks, which implies that any frontally induced

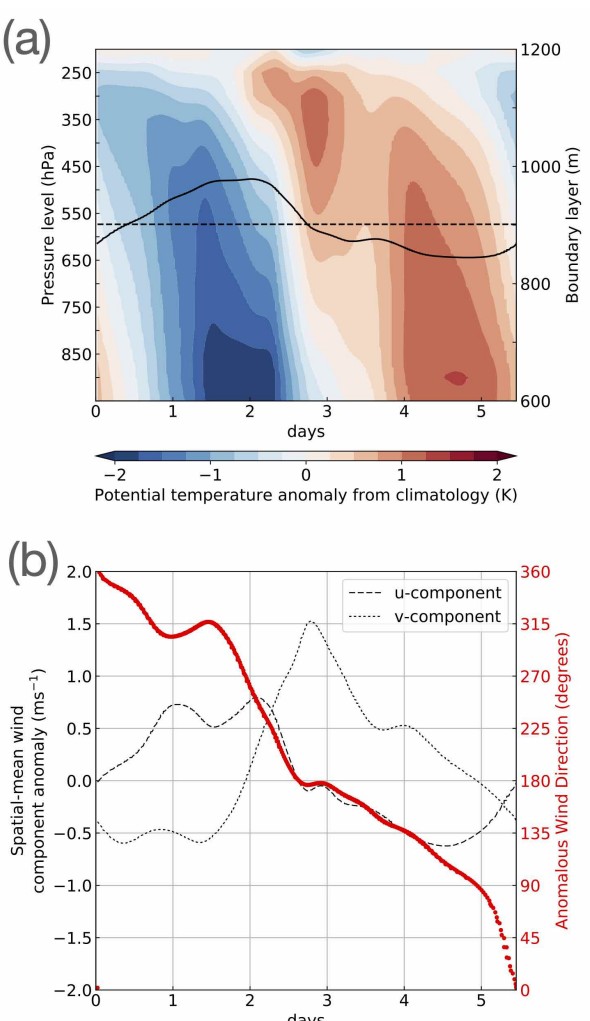

**Figure 10.** Phase tendency analysis along the closed trajectory shown in Fig. 9 for: (a) area-averaged potential temperature profile (colour shading, difference from winter climatology) and boundary layer height (dashed line, with winter climatological mean indicated by solid line); (b) meridional (dotted), zonal (dashed) anomalous wind components and corresponding anomalous wind direction (red dots) at the 950hPa level. The horizontal coordinate axis indicates the time progression in days along the closed trajectory.

temperature variance does not seem to dominate the signal, as the temperature variance generated by frontal circulations may

not co-vary with surface flux variance.

In light of this, we conclude that in the FT index growing phase the trigger for heat flux variability corresponds to the advection of (relatively) uniformly cold air masses over the spatially varying SSTs of the Gulf Stream extension region. The strong vertical contrast in temperatures causes enhanced surface heat fluxes which are then followed by a reaction in the lower

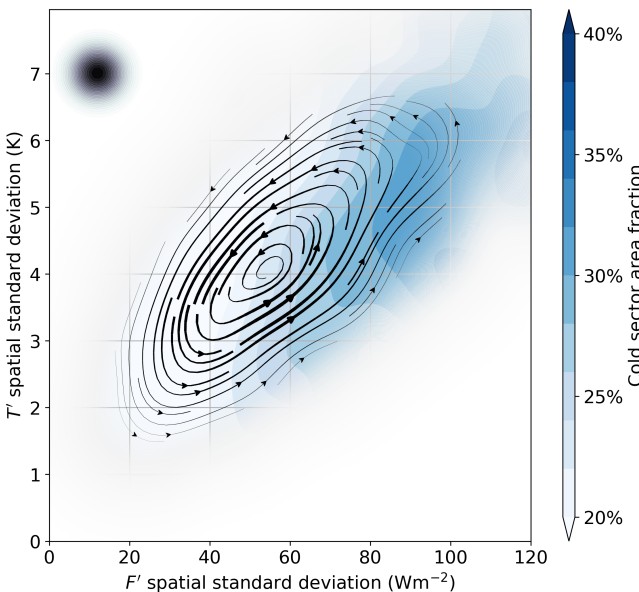

**Figure 11.** As in Fig. 8, for the phase tendency of cold sector area fraction (percentage of spatial domain occupied by cold sector). The size of the averaging Gaussian filter is indicated by the black-shaded dot in the upper-left corner.

atmosphere which experiences a subsequent increase in temperature spatial variability. Despite the SST field changing on much
longer timescales, a fixed SST front would therefore still induce heat flux spatial variance on synoptic timescales.

## 5  Conclusions

Lorenz (1955) showed that diabatic generation of available potential energy is proportional to the covariance between heating and air temperature. Globally, the stationary component of this term has been estimated to be positive as the residual of momentum and thermodynamic equations (Oort, 1964; Oort and Peixoto, 1974; Ulbrich and Speth, 1991; Li et al., 2007;
Marques et al., 2009), which suggests it acts as a source of energy for weather systems to feed on. A different picture is obtained for the transient component whose sign and magnitude has been observed to vary seasonally, with the strongest negative values occurring in the winter months (Ulbrich and Speth, 1991; Chang et al., 2002; Chang and Zurita-Gotor, 2007). This provides hints that, locally, surface heat fluxes behave overall as a sink of energy in the evolution of weather systems. Using data for surface heat fluxes and air temperatures from ERA-Interim, we find that they are locally negatively correlated
in time and space, in particular upstream of the N. Atlantic storm track, consistent with more recent literature.

In particular, we investigate the heat flux–temperature covariance through the definition of an index (FT index) that measures the local spatial covariance between sensible heat flux and air temperature at 850 hPa. To that effect, a hybrid approach was

taken where anomalies are defined as deviations from a spatial mean relative to a limited spatial domain, in our case, the Gulf Stream extension region.

The FT index is found to be always negative and characterised by bursts of activity coinciding with strong synoptic storm activity within the spatial domain considered. Composite analysis of strong index values suggest that heat flux-temperature spatial covariance behaves as an energy sink in the evolution of a storm. The peak of the FT index coincides with the onset of the decaying phase of the storm.

Heat flux-temperature spatial covariance, as measured by the FT index, and local baroclinic growth rate, as identified by
baroclinicity, are seen to be interacting in a cyclical evolution. Strong FT index values coincide with baroclinicity depletion, while a weaker FT index allows the baroclinicity to recover.

Spatial correlation and standard deviations in heat fluxes and air temperatures are observed to be equally important in the build up of strong spatial covariance, with an increase of spatial variability in surface heat fluxes typically preceding an increase for air temperatures spatial variability. In fact we find, rather counter-intuitively, that the correlation between flux and
temperature increases strongly with their variances.

We show that the intensification phase of the FT index coincides with the passage of a storm's cold sector across the region considered, which is compatible with the flux variance field shown in Fig. 1. The advection of cold air masses across the meridional SST gradient and mesoscale oceanic eddies then leads to an increased spatial variability in the surface heat flux field, which lead to the FT index to peak values, as heat flux and temperature fields correlate spatially.

Because the FT index is shown to be a good measure of baroclinicity depletion, and peak FT index values are dominated by cold-sector interaction with the spatial SST variance, our results show that the cold sector air-sea fluxes are a thermodynamic sink on the growth potential of storms.

*Data availability.* In this study we utilised data from ECMWF ERA-Interim, which is freely available from the ECMWF at https://apps. ecmwf.int/datasets/.

*Author contributions.* AM performed data analyses and prepared the manuscript. MHPA originated the idea for the study, contributed to the interpretation of the results and contributed to the manuscript. We are thankful to both reviewers for their insightful and detailed comments which helped improve the manuscript.

*Competing interests.* The authors declare that they have no conflict of interest.

*Acknowledgements.* AM acknowledges PhD studentship funding from the Scenario NERC Doctoral Training Partnership grant NE/R008868/1, co-sponsored by the UK Met Office. The authors thank ECMWF for making their ERA-Interim Reanalysis dataset freely and publicly available.

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
