# Peer review of "The role of heat flux-temperature covariance in the evolution of weather systems"

_Weather and Climate Dynamics, 2020_

## Referee Comment (RC1) · Anonymous Referee #1 · 29 Jun 2020

Review of "The role of heat flux-temperature covariance in the evolution of weather systems" by Marcheggiani and Ambaum

The authors discuss the covariation between air-sea heat fluxes and tropospheric temperature in the North Atlantic. At synoptic timescales, they find a negative covariance between these fields. They propose that air-sea fluxes on the cold sector of atmospheric storms are enhanced where the spatial variability of the SST is the strongest leading to a subsequent variability in the atmospheric temperature.

Although the paper presents original results about air-sea interactions, I am a bit skeptical in their significance. Also, I am concerned by different issues that would need to

be addressed before publication.

Recommendation: Major revision (or perhaps even reject in present form)

— One of the drawbacks of the paper concerns the physical meaning of the anomalies. The spatial variability of the sea surface temperature (SST) in the region near the Gulf Stream (GS) is generally due to the GS SST front. Here the spatial variability of the fluxes and temperature anomalies for timescales inferior to 10 days are not even presented. Is it related to the GS front? Centered over the front, or on its warm side? Is it related to oceanic eddies (as claimed near the end of the paper)? Understanding what are the characteristics of this variability is essential to the interpretation of the main results.

Another drawback is that a major process of air-sea interactions is completely over-looked: the so-called "oceanic baroclinic adjustment", as introduced by Nakamura et al. Their mechanism relies on the feedback of atmospheric temperature on air-sea fluxes. It seems to me that the results of the present manuscript are in disagreement with their findings. This issue should be tackled.

A last drawback relies in the motivation of the paper, i.e. the study of the generation/depletion of available potential energy (APE) by air-sea fluxes. Surface fluxes are involved in the budget of temperature inside the boundary layer, not in the 850hPa temperature budget. Hence the product air-sea heat flux times 850hPa temperature (above the MABL) cannot be interpreted as a term related to APE production. It is more simply related to the relation of air-sea fluxes with the free troposphere.

Detailed comments are given below.

— Major points:

1) I would have guessed that the variability of the air-sea fluxes lies above the warm side of the GS front, so that the spatial average used in (1) only captures this mode of variability. Indeed this seems to be the case when looking at Fig.3. But, this would

contradict line 257 where you say that spatial variability (at synoptic timescales) is due to mesoscale oceanic eddies.

Here are some different questions:

- Can you provide some information about the variability of SST (e.g. std(SST'))?

- Given the dataset you use ( ERA-I at 1.5deg of resolution), you are unable to represent the small spatial scales present in the fields you examine. I would like that you redo Figure 3 with a higher resolution dataset (e.g. ERA-5 at 0.25deg) to see more clearly whether the SST front is important or not.

- I do not see the point to show the SLP variance in Figure 1. Instead, could you present the std of F' and T' as well as the SST contours?

- Lines 212-213, You present a scenario where a cold front moves above a spatially varying SST, which would trigger spatially varying heat fluxes and then spatially varying T850. But, in my opinion, the cold front is already associated with a strong T850 anomaly. Please explain why and how this anomaly will be enhanced (in particular at what spatial scales).

- Can you show a figure of the time averages of [ F'* T'*] and [ F' T'] to contrast in which spatial region the synoptic eddies give a different response to the total eddy field?

2) You do not discuss at all of the mechanism proposed by H. Nakamura (Nakamura et al 2008 in GRL , Sampe et al. 2010 in J. Clim., Hotta and Nakamura 2011 in J. Clim), called the oceanic baroclinic adjustment (see Fig.12 in S10 or Fig. 20 in H&N11). This mechanism is related to a feedback between air-sea fluxes and surface temperature. Hotta and Nakamura relate the cold air advection of synoptic eddies to the interaction between air-sea temperature difference and air-sea fluxes. They stress the importance of SST gradient and surface baroclinicity. How does this relate to your Figure 4 and the co-evolution of T' and F'? More generally, please discuss their mechanism in comparison to yours.

3) I don't understand why you motivate your study by saying that FT is related to potential energy generation :

- Diabatic heating does not produce work, contrary to what is stated in line 2.

- Surface fluxes are only involved in the budget of temperature inside the boundary layer (see for instance Small et al. 2013 in Clim. Dyn.). Hence the product air-sea heat flux times 850hPa temperature has no physical meaning, per se, and cannot be related to APE production, contrary to what is stated in lines 70-71.

- I don't understand why flux-temperature covariance affects baroclinicity (line 160), or APE generation (which is quite different from the former).

4) I have some trouble to understand how you relate you covariance index to baroclinicity.

- Lines 158-159, you state that "baroclinicity was found to be depleted during extreme FT". However, from Fig.3b, it seems to me that baroclinicity is enhanced. Please explain.

- Also, you seem to relate mean baroclinicity (related to temperature gradients) and available potential energy (related to temperature anomalies), line 160. Please explain.

- Lines 252-253, you state that "air-sea exchanges drives the depletion of the baroclinicity over the domain". You seem to conclude this statement from the FT index life cycle which is not related to the baroclinicity.

5) You seem to think that surface air temperature would not react as 850hPa temperature when computing covariances. Could you compute pdfs like Fig.2 using surface temperature (either 2m or 10m) instead of 850hPa temperature? From that point of view, I would also like that you add the 2m temperature and the SST in Figure 3.

5) The argument about the triggering for heat flux variability (line 265) would need more firm bases. Could you complement Figure 8b with time evolutions of sea level pressure

and surface wind direction?

6) The pdf file is really too big (40mb). It made my printer crashed. I urge you to produce a much smaller size pdf.

— Minor points:

a) Figure 1 is too small when printed. Also, I don't understand why you chose to plot the SLP standard deviation. It would have been more logical to plot the T850 and the air sea-fluxes standard deviations (in blue and red) as well as the SST, since it is the subject of this paper.

b) Baroclinicity (line 154) should be defined.

c) Can you keep the spatial projections the same between figures: by choosing either the Conus representation (Fig.3) or the cylindrical one (Fig.1)
* * *

---

## Referee Comment (RC2) · Anonymous Referee #2 · 14 Jul 2020

In this study, the authors examine the co-variability of surface sensible heat flux and 850 hPa temperature, and found that they are generally negatively correlated. They constructed phase diagrams which suggests that most of the strongest interactions occur in the cold sector of the weather systems, with the flux increase leading the 850 temperature increase, and hypothesized that the spatial variability in the SST is important in increasing the spatial variability of the temperature field. I think the results are interesting. Nevertheless there are a number of issues that are not well explained and should be addressed in the revision.

Major comments: 1) The authors define the FT index by the spatial covariance be-

tween time anomalies in air-sea heat flux and 850 hPa temperature (equation 1). Why use both spatial and temporal deviations? Can the authors motivate this a bit better? How might an equation governing the APE, defined as both deviation from spatial and temporal mean, look like? The original definition by Lorenz is for spatial eddies (deviations from zonal mean). Later, Orlanski and Katzfey (1991) derived an alternative form for transient eddies (deviation from time mean). Perhaps the authors should provide reference for defining the APE as deviation from both time and spatial average? Defining the "energy" for a local region by subtracting the mean over that region is not necessarily useful due to the ambiguity of flux and conversion terms as Plumb (1983) pointed out.

References: Orlanski and Katzfey, 1991, JAS 48, 1972 Plumb, 1983, JAS 40, 1669

2) Related to the preceding point, to me, subtracting both the spatial and time mean makes it more difficult to visualize exactly how the passage of a system (e.g. a cold front) over the region would look like. Perhaps the authors should show figures corresponding to a time sequence of both the total fields and the eddy fields to make it easier for readers to understand some of the relationships found in this paper which seem to be a bit counter-intuitive.

3) A surprising result is that spatial variability in F' leads the spatial variability in T'. For weather systems of this time scale, one would expect that it is the atmospheric anomalies that force F', and thus it is, as the authors wrote, "counter-intuitive" (line 211). The authors explained that this "can be explained by the advection of the cold air mass, in the cold sector of a weather system, moving over a more spatially variable SST field such as that of the Gulf Stream extension. SST variability would trigger heat flux spatial variance which would then lead to temperature variance generation". I don't think I can understand this explanation. As the authors point out, F' nearly always damp T', and thus it is difficult to imagine how spatially varying flux, which acts to mostly damp the temperature anomaly, might give rise to increase in the temperature variance. Perhaps the authors could show some sequence of snapshots along the phase space

trajectory to show how this could happen and thus explain this "counter-intuitive" point better?

4) One speculation about F' leading T'. F' reacts to surface temperature anomalies. The surface front leads the 850 hPa front by some time, could this lead to some time lag between F' and T'? Fig. 8b apparently shows upper level temperature anomalies leading lower tropospheric anomalies, but this is for the large scale baroclinic wave in which T anomalies tilt eastward with height (e.g. Holton's text book; Lim and Wallace 1991). The variance increase likely corresponds more to the propagation of the front rather than the large scale temperature anomaly associated with the baroclinic wave? Can the authors show that this is not the case?

Reference: Lim and Wallace 1991, JAS, 48, 1718

5) Lines 251-254: Increasing F' followed by decreasing baroclinicity does not really imply that baroclinicity is depleted by the air-sea exchange. Baroclincity could be depleted by the growth of the baroclinic wave which occurs at the same time as the air-sea exchange is increasing. Causality cannot really be inferred when several things are occurring at about the same time, even with some slight lead-lag relationship.

6) Lines 59-61: There are "local" estimates using reanalysis data. For example, Chang et al (2002) showed that near surface sensible heat flux damps APE. See also Swanson and Pierrehumbert (1997) who also showed that 850 hPa temperature anomalies are strongly damped by surface fluxes over the ocean.

References: Chang, Lee, and Swanson, 2002: J. Climate, 15, 2163 Swanson and Pierrehumbert, 1997: JAS, 54, 1533

7) Lines 270-272: As pointed out above, Chang et al (2002) showed that latent heating (formation of cloud and precipitation in the warm sector) does generate APE, but near surface sensible heating damps APE. They also showed that over the Atlantic, the net effect is damping in winter but there are some regions where there is net generation.

8) In several places, the authors alleged to the importance of oceanic eddies (lines 91, 256, 258, 292). The data used is 1.5 degrees, and even the full resolution of ERA-Interim cannot really resolve oceanic eddies. If oceanic eddies are so important then how could the analysis based on ERA-Interim reveal that?

9) The figures need to be improved. The legends are really small and can't be clearly seen without enlarging the figures by a lot.

Minor comments:

i) Line 156: "lies almost entirely on the negative side of the FT index". I thought the FT index is always negative (line 97)?

ii) Line 237: "A downward propagation of the temperature anomalies" - this is not really "propagation" - related to the eastward tilt of temperature with height in medium scale baroclinic waves discussed above.

---

## Author Comment (AC1) · 15 Aug 2020

The comment was uploaded in the form of a supplement:
https://wcd.copernicus.org/preprints/wcd-2020-19/wcd-2020-19-AC1-supplement.pdf
* * *

---

## Author Response (AR1)

**The role of heat flux-temperature covariance in the evolution of weather systems**

Marcheggiani and Ambaum

**Author response to reviewers**

We truly appreciate and are thankful for the effort that has been put by the Reviewers in reviewing our manuscript. Their comments are thoughtful and insightful and in responding to them we believe that the manuscript can benefit substantially. We hope that all their concerns have been duly addressed.

Reviewer's comments are reprinted in a *thinner and italicised style*, our response is typed below it in a thicker and non-italicised style.

Figures from the original manuscript are referred to following the manuscript's order while new figures included in this document are labelled as Figure AR# (Author Response).

**Author response to Reviewer #1**

We would like to thank Reviewer 1 for the detailed and insightful comments they gave on the manuscript, highlighting some unclear passages in the manuscript and highlighting studies on air—sea interaction which we did not discuss in our initial manuscript. Below we give a point-by-point response to the issues raised by the Reviewer.

One of the drawbacks of the paper concerns the physical meaning of the anomalies. The spatial variability of the sea surface temperature (SST) in the region near the Gulf Stream (GS) is generally due to the GS SST front. Here the spatial variability of the fluxes and temperature anomalies for timescales inferior to 10 days are not even presented. Is it related to the GS front? Centered over the front, or on its warm side? Is it related to oceanic eddies (as claimed near the end of the paper)? Understanding what are the characteristics of this variability is essential to the interpretation of the main results.

We believe there may have been some misunderstanding arising from our description of the mixed time-space anomalies. These are built as the spatial covariance of the departures from a 10-day running mean (which corresponds to a high-pass filter in the frequency domain), thus obtaining an instantaneous description of spatial patterns of synoptic-timescale variability (i.e. 10 days and below), which is what the paper is about. By removing a 10-day running mean, we are filtering out lower-frequency variability, such as seasonal variations, which may otherwise dominate the spatial variance, and which describes different physical processes.

We also emphasise that the high-pass time filtering occurs on the fluxes and the atmospheric temperatures, not on the SSTs. This means that the spatial variability caused by eddies on the gulf stream temperature front interacting with synoptic weather systems are in fact represented in our covariance index, and we show evidence that in the initial stage of development this is the key source of covariance.

We will rephrase the relevant passages in the manuscript to make this all more explicit and clear.

Another drawback is that a major process of air-sea interactions is completely overlooked: the so-called "oceanic baroclinic adjustment", as introduced by Nakamura et al. Their mechanism relies on the feedback of atmospheric temperature on air-sea fluxes. It seems to me that the results of the present manuscript are in disagreement with their findings. This issue should be tackled.

We are very thankful to the reviewer for highlighting the studies by Nakamura & co-authors on the role of the oceanic temperature front in storm track dynamics. In particular, they highlight the importance played by SST fronts in forcing a surface air temperature gradients through differential sensible heating across the SST front. This was shown to be essential for the maintenance of strong near-surface baroclinicity, which anchors the climatological storm track.

Our study does not contradict these results; in fact they are consistent with each other as well as complementary to each other. We find that the spatial variance of the fluxes, indeed

including contributions of the N-S gradient of SSTs over the oceanic front, are associated with instantaneous depletion of baroclinicity. This is consistent with the mechanism discussed in a series of papers by Ambaum & co-workers highlighting the role that eddies play in temporarily depleting the baroclinicity in a predator—prey like relationship; this relationship is really a familiar instance of the nonlinear life-cycle of midlatitude eddies where meridional heat fluxes locally deplete the meridional temperature gradient in the atmosphere; in the older literature this quasi-periodic predator—prey relationship would have been described as an index cycle. However, this does not contradict the fact that high eddy activity on average must be geographically associated with high baroclinicity, which is essentially what the Nakamura papers are about, and of course also classical papers, such as Hoskins & Valdes (1990), and also Ambaum & Novak 2014.

So on a synoptic time-scale baroclinicity is depleted by synoptically induced variance of fluxes, as the many diagnostics in our manuscript show in various ways, but climatologically of course the storm track is anchored geographically by the high temperature gradients in the oceanic front, as highlighted by processes elucidated in the papers referred to by the reviewer, and in further detail in Swanson & Pierrehumbert 1997.

It is clear that this mutually complementary but consistent view on baroclinicity and spatial SST variance is an angle which we did not at all highlight in any detail in the manuscript, and we will discuss this in much greater detail in the revision.

A last drawback relies in the motivation of the paper, i.e. the study of the generation/depletion of available potential energy (APE) by air-sea fluxes. Surface fluxes are involved in the budget of temperature inside the boundary layer, not in the 850hPa temperature budget. Hence the product air-sea heat flux times 850hPa temperature (above the MABL) cannot be interpreted as a term related to APE production. It is more simply related to the relation of air-sea fluxes with the free troposphere.

We agree with the reviewer that our index is not formally equivalent to a term in the APE production budget —indeed, we never claimed as much— and we will rewrite the manuscript to make that clearer than we managed to do in the first version. We acknowledge that the hybrid framework we use can lead to confusion and we will rephrase in a clearer way the reasons behind its use in our study.

In our paper we work towards a hybrid understanding of how APE can be affected locally, in particular in response to coupling of the free troposphere with the surface. As the reviewer pointed out, the intensity and sign of surface heat fluxes are typically computed from the energy budget at the surface, hence their covariation with higher layers of the atmosphere is not trivial, and we believe it can have an effect on the evolution of weather systems.

More informally, we examine how synoptic heat fluxes contribute to enhancing or depleting the local synoptic variance in the lower tropospheric temperature field. This local temperature variance is of course part of the global APE integral in the standard Lorenz energy cycle.

1) I would have guessed that the variability of the air-sea fluxes lies above the warm side of the GS front, so that the spatial average used in (1) only captures this mode of variability. Indeed this

seems to be the case when looking at Fig.3. But, this would contradict line 257 where you say that spatial variability (at synoptic timescales) is due to mesoscale oceanic eddies.

We look at the spatial covariance between heat flux and temperature time fluctuations from a centred ten-day average thus we expect part of the total variability over the Gulf Stream extension also to derive from the interaction between the atmosphere and colder waters. In particular, we surmised that the higher level of variability observed over the western end of the North Atlantic could be traced back to the presence of mesoscale oceanic eddies, which would provide for stronger spatial SST contrasts, and hence flux variance. In fact, in the eastern North Atlantic spatial variances and covariance between heat flux and temperature is much weaker (though we did not include this in the manuscript).

Here are some different questions:

- Can you provide some information about the variability of SST (e.g. std(SST'))?

SST time anomalies, defined as those used in the computation of the spatial covariance, would be much weaker than those for air temperature or heat fluxes as the former vary on much longer time scales. We point out again that we do not analyse the high-pass filtered SSTs, but the high-pass filtered fluxes. These contain the spatial variance introduced by the SST spatial variance, even if that had been chosen fixed in time.

The standard deviation in time of the SST is simply not a relevant diagnostic for pointing out a source of spatial variance in the synoptic time scale fluxes. A fixed SST front with no temporal standard deviation would still induce the spatial variance on synoptic time scales of the fluxes, which we diagnose. We will further emphasise this property in the revision to make this point clearer.

- Given the dataset you use (ERA-I at 1.5deg of resolution), you are unable to represent the small spatial scales present in the fields you examine. I would like that you redo Figure 3 with a higher resolution dataset (e.g. ERA-5 at 0.25deg) to see more clearly whether the SST front is important or not.

While it is true that ERA-I at 1.5°x1.5° resolution does not capture the smaller spatial scales, we found that using ERA-5 leads to slightly larger values for spatial covariance, as these smaller scales add to the variance. However, in the construction of composites, finer spatial details are lost due to the large number of events involved in the averaging process. Therefore, we believe the use of higher-resolution data proves most beneficial when looking at individual case studies.

- I do not see the point to show the SLP variance in Figure 1. Instead, could you present the std of F' and T' as well as the SST contours?

We agree that SLP variance is not the best choice in this context and we modified it accordingly (see Fig. AR1), thanks for pointing this out.

It is interesting to note that the peak flux standard deviation has a small bias towards the southern side of the SST front confirming a previous point by the Reviewer. This pattern is

completely consistent with the mechanism of cold sector being advected in the SE direction over warmer (and spatially variable) SSTs.

- Lines 212-213, You present a scenario where a cold front moves above a spatially varying SST, which would trigger spatially varying heat fluxes and then spatially varying T850. But, in my opinion, the cold front is already associated with a strong T850 anomaly. Please explain why and how this anomaly will be enhanced (in particular at what spatial scales).

What is enhanced is the spatial variance of surface heat fluxes which is then followed in time by an increase in temperature spatial variance (we do not imply a causal link, which would act opposite). When the cold front moves across the spatial domain, the temperature spatial variance does not change significantly, while the surface heat fluxes pick up spatial variance when the cold sector moves over warmer SSTs. This is exactly the type of processes that our diagnostics highlight. Note also that this process is consistent with the new Figure 1 (Fig. AR1 in this response).

Figure AR1: Update of Figure 1 from manuscript; shading represents temporal standard deviation of F, contours represent SST winter climatology (every 2K from 280K to 290K, every 5K otherwise).

- Can you show a . of the time averages of [ F'\* T'\*] and [ F' T'] to contrast in which spatial region the synoptic eddies give a different response to the total eddy field?

The time average of either [F'\*T'\*] or [F'T'], assuming the brackets to be indicating a spatial average operator, would correspond to a negative number rather than a field. The time average of F'\*T'\* would instead provide a picture of where the spatial covariance of heat flux and temperature is realised within the spatial domain we considered. This is found to peak along the Gulf Stream, where time variances of the fluxes are also larger (see Fig. AR2 - and compare to Fig. AR1).

Thank you for pointing out this useful diagnostic; we will describe this property in the revision.

Figure AR2: Wintertime (DJF, 1979-2019) mean (shading) of product between time-space anomalies in flux and temperature over the spatial domain selected for our study. Black contours represent wintertime SST climatology (every 2K from 280K to 290K, every 5K otherwise).

2) You do not discuss at all of the mechanism proposed by H. Nakamura (Nakamura et al 2008 in GRL, Sampe et al. 2010 in J. Clim., Hotta and Nakamura 2011 in J. Clim), called the oceanic baroclinic adjustment (see Fig.12 in S10 or Fig. 20 in H&N11). This mechanism is related to a feedback between air-sea fluxes and surface temperature. Hotta and Nakamura relate the cold air advection of synoptic eddies to the interaction between air-sea temperature difference and air-sea fluxes. They stress the importance of SST gradient and surface baroclinicity. How does this relate to your Figure 4 and the co-evolution of T' and F'? More generally, please discuss their mechanism in comparison to yours.

We replied to this earlier and we agree that it will be a valuable addition to the manuscript to discuss the relation between these arguments and ours. As indicated before, we do not think they are contradictory at all; rather, they are complementary and really speak of different properties of the storm track.

3) I don't understand why you motivate your study by saying that FT is related to potential energy generation :

- Diabatic heating does not produce work, contrary to what is stated in line 2.

Thank you for pointing this out. That was indeed poorly phrased and will be changed. What we of course meant to say is that local diabatic heating and temperature anomaly fields need to be positively correlated for diabatic heating to maintain a circulation against dissipation.

- Surface fluxes are only involved in the budget of temperature inside the boundary layer (see for instance Small et al. 2013 in Clim. Dyn.). Hence the product air-sea heat flux times 850hPa temperature has no physical meaning, per se, and cannot be related to APE production, contrary to what is stated in lines 70-71.

- I don't understand why flux-temperature covariance affects baroclinicity (line 160), or APE generation (which is quite different from the former).

We are aware that the link we made between baroclinicity and available potential energy is informal and that direct expressions for local APE have been devised (Novak and Tailleux, 2017). As pointed out in an earlier response, we will put more effort in rephrasing clearly our intentions and the reasons why we have been using this particular framework.

4) I have some trouble to understand how you relate you covariance index to baroclinicity.
- Lines 158-159, you state that "baroclinicity was found to be depleted during extreme FT".
However, from Fig.3b, it seems to me that baroclinicity is enhanced. Please explain.

Composites shown on the left in Figure 3 are relative to the peak in the covariance and we did not include lagged composites for the sake of conciseness. Perhaps it is useful to add composites at negative lags to illustrate more clearly how baroclinicity varies, as its depletion at larger FT covariance was indicated by phase tendencies presented in a later section.

In Fig. AR5 in this document it can be seen that the near surface temperature (T2m) at the peak does not particularly appear to coincide with an enhanced N—S temperature gradient. Furthermore, we produced a new Figure 3 (Fig. AR8 in this document) where it can be seen that the overall N—S gradient in T850 does not enhance at the peak of the index although local T850 gradients do appear to be somewhat enhanced, as suggested by the Reviewer. We will discuss this in the revision.

---

## Author Response (AR2)

**The role of heat flux-temperature covariance in the evolution of weather systems**

Marcheggiani and Ambaum

**Author response to reviewers/editor**

We would like to express again our gratitude towards both reviewers for the time and effort they put in reviewing our work. We are also thankful to the editor for managing the reviewing process. We hope this second round of revisions will fully address any remaining issues.

Reviewer's comments are reprinted in a *thinner and italicised style*, our response is typed below it in a non-italicised style.

**Author response to the Editor's comment:**

Consider adding your thoughts on the role of vertical motion and moist diabatic processes as potential drivers of the spatio-temporal variance of T' at the 850 hPa. In your discussion of F'T' much attention is given to horizontal air mass transport. However, along the fronts of a growing baroclininc wave, we must expect vertical motion to be strong and adiabatic and diabatic modification to affect T' at 850 hPa and consequently F'T'.

We agree that one might expect vertical motions associated with along-frontal flow to play a significant role in driving the FT index. Our data indicate that this appears not to be the case, at least not dominantly so.

If the index would peak when frontal circulation is strongest we would expect the mean temperature to be around average, as a front is associated with both anomalously warm and cold air masses. Instead, we find that the index peaks when the area mean temperature is coldest, and when the cold sector area is largest. So although we do not rule out a role for frontal diabatic effects on driving the FT index, we think that the data are more consistent with cold sector processes.

A reason may be simply that the temperature variance that is generated by frontal circulations does not co-vary with surface flux variance; indeed this may well be expected to be the case. However, we also find that temperature variance peaks when the index (i.e. FT co-variance) peaks, which means, again, that any frontally induced temperature variance does not seem to dominate the signal.

We have decided to add this discussion in the text towards the end of Section 4.

**Author response to Reviewer #1**

The authors have responded to my comments and added relevant figures and discussion. I have still some minor comments before the paper to be published.

My recommendation is Minor revision

1) Lagged composites (as discussed on lines 157-166) showing F' and T' (or other quantities) might be insightful to better understand the counter-intuitive results of your study. Maybe it would be useful to include them.

The lagged composites show the temporal evolution of the system presented in Fig.3b, which represents the peak in its evolution. However, a large number of plots would need to be appended to fully describe this process and we found that it does not add any relevant information to the summary we provide in lines 157-166. We have of course done much more analysis than that presented in the paper, but we have to be selective in what we present.

2) Discussion lines 189-200 and related Figure 5.
It seems to me, by looking at the contours and the arrows, that when the FT index is strengthening from -100 to -400Wm-2K, baroclinicity stays the same, around 0.6day-1, contrary to what you state line 194.
The change of baroclinicity would depend on the initial value of it?
My problem may be due that you change the figure with the revision.

We apologise for the confusion that our original wording lead to. We actually meant that baroclinicity begins to decrease once the strengthening of the FT index has occurred. The figure has changed as we modified the limits of the plotting region in order to concentrate more on where data is present and that might have stretched the original picture. We would also like to stress the fact that where the data distribution is higher baroclinicity is actually observed to decrease slightly with a strengthening FT index.

We have rephrased that passage in order to avoid confusion.

3) You should check the paper of Dacre et al. (in WCD, 2020) who examined cyclone-relative heat fluxes. This may be relevant to your study.

We thank the reviewer for the suggestion, we had the chance of speak about this study with one of the authors (when in-person speaking was still allowed!). In some sense their study is opposite to ours as it investigates the effect of heat fluxes on SST anomalies but it is definitely of interest and relevant to what we are currently working on.

4) Line 201. You discuss the findings of Hotta and Nakamura, without saying what they are.

Indeed, our paragraph only implicitly describes their results by showing how our work is complementary. We have now more explicitly spelled out the findings we were referring to.

**5) Line 291. Do you refer to surface 10m or 850hPa winds?**

Apologies for not mentioning explicitly that we refer to 950hPa winds, as we did not have data readily available for 10m winds at the time the revision was being carried out. We did check that we get a similar picture for winds at 10m. We added this detail to the text.

6) The first sentence of the introduction is awkward:

... \*\*storms tracks\*\* are longitudinally localized, with the main regions of intense \*\*storm track\*\* activity ...

Again, Line 81 -> Given that \*\*storm tracks\*\* are by definition the main reservoirs of eddy potential energy...

I am confused. Do you refer to the same entity? i.e. paths of individual storm and regions where the eddy kinetic energy or potential energy is high? You should be consistent throughout the paper: storm tracks should be defined from the start.

In our paper, we refer to storm tracks as those regions where storm activity is most intense rather than the individual tracks of storms. There are many ways of defining a storm track region, either from Lagrangian or Eulerian perspectives, which give slightly different pictures but essentially the same in substance and we mention storm tracks mainly to contextualise our choice for the spatial domain over which we calculate the spatial covariance. We rephrased that first sentence to make it clearer.

**Author response to Reviewer #2**

I appreciate the authors' effort in substantially revising this manuscript. In my opinion, the discussions have improved. Nevertheless, there are still some issues that the authors should address before the paper can be accepted.

1) My main remaining issue is the authors' contention that the air-sea fluxes act to decrease baroclinicity (lines 189-200). I pointed out in my previous review that the fact that baroclinicity decreases while the FT increases does not imply the flux causes decrease in baroclinicity, as the synoptic eddy is growing when FT increases, and the synoptic eddy growth could be the cause of the decrease in baroclinicity. In the response and in the revised manuscript, the authors acknowledged that they can't separate the effect of the heat flux from the effect of the eddy growth in reducing baroclinicity. Nevertheless, they still contended that since the air-sea flux damps the temperature variance, this provide strong evidence for the eddy flux damping baroclinicity. I don't really follow this point how the air-sea flux damping eddy amplitude would imply it would erode the MEAN temperature gradient. This needs to be better explained. This also relates to the discussions in the paragraph on lines 210-213. I can see how the air-sea flux is important in the budget of eddy APE, but I can't really see any "strong" evidence (apart from Fig. 5, which I have argued that this does not show causality) that these fluxes deplete the mean baroclinicity.

We observe mean baroclinicity to be depleted only after heat flux—temperature covariance reaches its peak value. These peak values would correspond typically with baroclinic waves in their most active phase during which mean baroclinicity is destroyed. This decrease in baroclinicity could be said to be due primarily to the kinematic action of eddies in transporting air masses across the large scale horizontal temperature gradient and FT covariance might just happen to be stronger during this stage of the eddy evolution thus only coincidental.

At low frequencies, the air—sea heat fluxes of course maintain and anchor the high baroclinicity regions (we now emphasize this in the revision with references, amongst others, to Hotta & Nakamura). But at high frequency we observe a different phenomenon: the damping effect quantified by negative F'T' covariance corresponds to a reduction in baroclinicity at those short timescales. This is what can be observed in Figure 5. Note that the figure shows the high frequency anomalies in the baroclinicity. Although we agree that descriptive statistics can only ever be descriptive, and cannot settle causality, the data are consistent with the physical picture we paint in the paper.

We certainly concede that we should not use words like "strong evidence".

**We have revised the wording to make the above clear**

**Other comments:**

2) Lines 32-33: Note that as I pointed out in my previous review, Chang et al (2002) showed that total diabatic generation of transient EPE is largely negative except over parts of the Pacific, with the damping effect of sensible heating largely cancelling the positive generating due to latent heating. In fact, Chang and Zurita-Gotor (2007, JAS 64, 2309, see their table 1) have shown that over the northern hemisphere mid latitudes, and especially in the Atlantic, diabatic heating damps transient eddies. These results do not contradict those cited in this study (Li et al, 2007;

Marques et al., 2009), because in Lorenz's definition of EAPE, eddies are defined as deviations from the zonal mean, and includes stationary eddies. In winter, net diabatic heating is positive over the warm oceans, and negative over the cold continents (e.g. Held et al., 2002, J. Climate, 15, 2125, Fig. 8), leading to generation of EAPE but mainly for stationary eddies and not necessarily for transient eddies that form the storms. My point is that there exists previous studies that showed that diabatic heating acts to damp baroclinic waves instead of being a source of energy in storm development (lines 33, 337-338).

We are thankful to the reviewer for pointing this out. We agree that those studies should be mentioned in the relevant section and we changed the text accordingly. We did actually find a similar seasonality in the FT index to that observed in Chang and Zurita-Gotor (2007), as the FT index weakens dramatically over the summer months, and this reference might reveal useful for future work.

3) Equation 1. Here the authors should discuss why they used 850 hPa T' rather than 2m T'. This is discussed below (starting line 148) but in my opinion should be discussed here instead.

We changed the text and referred the reader to the discussion presented later which we believe more logical to keep in Section 3, as we would need to refer to figures not yet presented when introducing equation 1.

4) Lines 112-114: I still don't understand how the effects of ocean eddies on F at the resolved scales would be capture by the reanalysis system. As far as I know, F is model generated, hence if the model does not resolve the ocean eddies in the SST, how can F contain the effects of that? You could argue that the effects of the ocean eddies on the resolved scale temperature structure may be captured since temperature is an observed input that is assimilated, but I am not convinced that the effects of ocean eddies on F can be captured by the model. Either remove this discussion or explain more clearly.

The computation of F in the reanalysis system relies strongly on the air temperature at the surface, whose variability includes the influence of oceanic eddies. In this way, the model would be able to capture part of the effects of oceanic eddies on F. We agree with the reviewer that higher resolution may prove beneficial, however, this is the best spatial resolution available for reanalyses datasets. We added to this caveat in the revised manuscript.

5) Line 127: The heat flux anomalies seems to be skewed towards positive values to me?

Indeed, we changed negative to positive. We were initially referring to the index values distribution which has an extended negative tail and mixed up the two distributions, though the argument we present is made on the basis of the distribution for F'\* being skewed towards positive values. Apologies for the confusion.

6) Lines 155-156: Perhaps the near surface temperature anomalies are more strongly damped by the underlying SST may also contribute?

Indeed, this is something we implied in lines 149–151. It is true that high frequency temperature anomaly values are weaker near the surface. We added an explicit reference to that in the revision.

7) Line 162: "derives from a gain/loss of signal due to averaging" - perhaps add something like "partially"? Not clear to me that all of the signal is due to that as the authors also argued below.

**Agreed - we rephrased that sentence accordingly.**

8) Lines 174-175: I don't quite understand the phrase "it is reasonable to interpret any positive instances or moderately negative values as indicative of a relatively weak heat exchange" means.

When the spatial covariance between F' and T' is close to zero is associated with low F' and T' variances or very weak spatial correlation. Figure 7 further excludes the possibility of high variances at low correlation, therefore we expect low covariance to be associated with a relatively calmer weather activity and thus weaker heat exchanges.

9) Fig. 3 shows that the covariance between F and T is positive along the Canadian coast. Can the authors explain that?

We believe the reviewer is actually referring to the positive values in time mean F'\*T'\* shown in Fig. 4, as Fig. 3 does not contain plots of covariance. The positive instances occur over regions that are frequently covered by sea ice which means the ocean there has a different heat capacity and negative (positive) temperature anomalies could be collocated with heat fluxes into (out of) the ocean linked with ice formation (melting).

10) Line 182: phases should be phase?

Yes, it should, thank you for spotting that.

11) Lines 189-190: The Eady growth rate maximum should probably be defined?

We utilise the same definition given in Hoskins and Valdes (1990) and, similarly to Ambaum and Novak (2014), we calculate for 750hPa level using centred-difference approximation for the zonal wind vertical gradient between 650 hPa and 850hPa. We specified this in the revised manuscript.

12) Line 196: "to reduce" should be changed to "to decrease"

**We changed that.**

13) Line 254: "consistently" should be "consistent"?

**We changed that.**

14) Line 317: As discussed in item 1), I'm not convinced that this is the case.

As discussed above, we hope this is now clearer.

[revised manuscript text omitted]